# Characteristics and Projection of Rainfall Erosivity Distribution in the Hengduan Mountains

Xinlan Liang [1,†], Lei Zhang [1,†], Shuqin He [2], Ke Song [1] and Zicheng Zheng [3,*]

[1] College of Water Conservancy and Hydropower Engineering, Sichuan Agricultural University, Ya'an 625014, China; liangxinlan@sicau.edu.cn (X.L.); 2021220005@stu.sicau.edu.cn (L.Z.); songke@stu.sicau.edu.cn (K.S.)
[2] College of Forestry, Sichuan Agricultural University, Chengdu 611130, China; hesq@sicau.edu.cn
[3] College of Resources, Sichuan Agricultural University, Chengdu 611130, China
[*] Correspondence: zichengzheng@aliyun.com
[†] These authors contributed equally to this work.

**Abstract:** This study examines the spatiotemporal variations of rainfall erosivity in the Hengduan Mountains, known for their rugged terrain and high potential for soil erosion risks, over the past 30 years. Additionally, it investigates the changing trends of rainfall erosivity between 2025 and 2040 under the Sustainable Development Pathway 2–4.5 (SSP2–4.5), using four Global Climate Models (GCMs) based on the Coupled Model Intercomparison Project phase 6 (CMIP6). The results indicate: (1) The annual distribution of rainfall erosivity in the Hengduan Mountains exhibited significant seasonal variations, ranking in the order of summer > autumn > spring > winter on a seasonal scale. (2) Over the past 30 years, there has been a slight decrease in annual precipitation and a corresponding slight increase in rainfall erosivity. Periodic extreme values occur every 6–8 years. (3) Spatially, rainfall erosivity demonstrates a decreasing gradient from southeast to northwest. There is a significant positive correlation between rainfall erosivity and precipitation, while a significant negative correlation exists with elevation in the vertical direction. Furthermore, the northeastern part of the Hengduan Mountains exhibits an increasing trend of rainfall erosivity, while the southern region experiences a decreasing trend. (4) Considering the joint driving forces of increased precipitation and erosive rainfall events, rainfall erosivity is expected to significantly increase in the future, posing a more severe risk of soil erosion in this region.

**Keywords:** Hengduan mountains; rainfall erosivity; distribution; projection

## 1. Introduction

In recent years, soil erosion has become an increasingly severe issue due to the influence of global climate change. This phenomenon leads to global soil depletion, wastage of water resources, farmland degradation, and ecosystem destruction [1]. Rainfall plays a crucial role in soil erosion, and rainfall erosivity serves as an important indicator for assessing the extent and risk of soil erosion [2]. Rainfall erosivity refers to the impact energy generated by raindrop impact on the surface, which is the primary driving force behind the detachment and transport of soil particles [3]. Intense rainfall results in higher erosivity, accelerating the soil erosion process. Various factors influence rainfall erosivity, including rainfall intensity, rainfall amount, raindrop size, and raindrop velocity [4].

Research on rainfall erosivity in soil erosion mainly focuses on two aspects. Firstly, by measuring and analyzing rainfall erosivity, we can quantitatively assess the impact of rainfall events on regional soil erosion in terms of time and space, providing a scientific basis for soil conservation and land management [5,6]. Secondly, analyzing and studying rainfall erosivity enable us to predict future trends in soil erosion. Long-term observation and analysis of erosivity from different rainfall events allow us to understand the development trends of global or regional soil erosion against the background of climate change. Based

on different local conditions and climate scenarios, some regions exhibit an increasing trend in rainfall erosivity [7,8], while others show a decreasing [9,10] or significantly decreasing trend [11]. In certain areas, the number of erosive rainfall events decreases, but their intensity increases, leading to complex variations in rainfall erosivity within those regions [10]. This indicates that the development trends of soil erosion in different regions worldwide are not consistent. To help us take appropriate measures to mitigate the risk of soil erosion and ensure the sustainable utilization of land resources, it is crucial to consider the information on regional soil erosion trends provided by rainfall erosivity in future soil management and agricultural practices.

Research on rainfall erosivity primarily focused on the construction and modeling of rainfall erosivity datasets. A global rainfall erosivity database has been established by integrating remote sensing databases, measurement data, and climate datasets [12,13]. This database has been effectively applied in various studies [14,15]. In terms of modeling, scholars have developed rainfall erosivity calculation equations based on daily, monthly, seasonal, and annual rainfall data by investigating the mathematical relationship between $EI_{30}$ and rainfall amount [16–18]. The semi-empirical model proposed by Chinese scholar Zhang Wenbo [19], known as the "semi-monthly" model, has been validated as having a low error rate in calculating rainfall erosivity in China. Many researchers have employed this model to study rainfall erosivity in different regions of China [20,21], making it particularly suitable for the precipitation-rich Hengduan Mountain region.

Previous studies have assessed the extent and trends of soil erosion, predicted future changes, and formed a mature research system for rainfall erosivity by investigating the global and regional situation. Rainfall erosivity density is the ratio of rainfall erosivity and precipitation [22], reflecting the rainfall characteristics that cause regional rainfall erosivity, such as rainfall intensity and duration. It is widely used in rainfall erosivity-related research [23,24]. Linear regression analysis is effective in reflecting the direction of long-term climate factor changes and has been successfully applied in many regions of the world [23,25]. Theil–Sen trend analysis, as a non-parametric trend estimation method, has high computational efficiency, is not affected by outliers, and is more suitable for studying climate phenomena [26]. The Mann–Kendall trend test method is commonly used in the trend study of rainfall erosivity [6,27]. The combination of these two methods greatly enhances the reliability of trend analysis results [26]. In terms of rainfall erosivity prediction, related research has successfully applied the CMIP6 model to predict regional rainfall erosivity [28,29]. These research results generally indicate that, under the risk of increasing rainfall erosivity in the future, soil erosion problems will be exacerbated. These research achievements have validated the feasibility of using the CMIP6 to predict rainfall erosivity in this study.

CMIP6 is the latest stage of the World Climate Research Program (WCRP), with significant improvements in both spatial accuracy and simulation process compared to Coupled Model Intercomparison Project phase 5 (CMIP5) [30]. The CMIP6 model considers different emissions scenarios for various socioeconomic pathways (SSPs) and quantitatively predicts future climate based on temperature, precipitation, solar radiation, and land use [31]. The description of SSPs for future socioeconomic evolution has a clear advantage in modeling and evaluating the ecological status of the natural environment [32]. Therefore, given the context of global climate change, predicting rainfall erosivity in the Hengduan Mountains based on CMIP6 model is crucial for future land resource management, ecological protection, and sustainable development in the region.

The Hengduan Mountains are situated in the transitional zone between the Qinghai-Tibet Plateau and the plains. They were formed due to the compression and collision of the Indian Plate subducting beneath the Eurasian Plate. This geological process resulted in a series of north–south geological folds, which is distinct from the predominantly east–west orientation of most other mountain ranges in China. Additionally, the deep incision by rivers has created high mountains and deep valleys in the Hengduan Mountains, leading to pronounced vertical zonation of climate [33], vegetation [34], and erosion characteristics [35].

The rugged terrain and abundant precipitation in the region contribute to a higher degree of erosion compared to the national average [36]. Consequently, the area experiences widespread development of gullies and ravines, severe soil degradation, and frequent geological disasters such as landslides and debris flows [34]. However, there have been limited reports exploring regions with such pronounced topographic variations and distinct vertical zonation of erosion characteristics. Understanding the spatiotemporal variation patterns of rainfall erosivity in this region is of great significance for assessing potential soil erosion risks in the region.

Therefore, this study focuses on two main aspects. Firstly, it employed trend analysis methods to examine the temporal variation trend of rainfall erosivity. It also simulated the spatial distribution patterns of rainfall erosivity using spatial interpolation methods and explored the relationship between rainfall erosivity and precipitation at different altitudes. These analyses aim to reveal the spatiotemporal differentiation patterns of rainfall erosivity, assess potential soil erosion risk zones, and identify key areas for rainfall erosivity control in the region.

Secondly, utilizing the Delta downscaling method, it extracted data from five atmospheric circulation models based on CMIP6 to predict the trend of rainfall erosivity in the Hengduan Mountains from 2025 to 2040. This endeavor seeks to provide reference information for long-term soil and water conservation efforts in the region.

## 2. Materials and Methods

### 2.1. Overview of the Study Area

The Hengduan Mountains region is a transitional zone between the Qinghai-Tibet Plateau, the Sichuan Basin, and the Yungui Plateau. It is located approximately between 24.5° N to 33.9° N latitude and 96.3° E to 104.5° E longitude (Figure 1). The elevation ranges from 600 to 7200 m, gradually decreasing from north to south and from west to east, encompassing a total area of approximately 360,000 square kilometers [37]. The Hengduan Mountains region exhibits complex terrain, characterized by high mountains and deep valleys, making it one of the most climatically diverse vertical zones in the world. The region receives winter and spring precipitation through westerly wind moisture transport, while summer and autumn precipitation are influenced by monsoon transport [38]. These distinct moisture sources result in uneven precipitation distribution throughout the year, with well-defined wet and dry seasons. The north-south-oriented mountain ranges impede the westward transport of moisture, leading to moisture accumulation in the eastern and southern parts of the Hengduan Mountains. Conversely, the western areas experience longer sunshine duration, dry air, and reduced precipitation. Overall, the distribution of precipitation exhibits a decreasing trend from south to north within the study area.

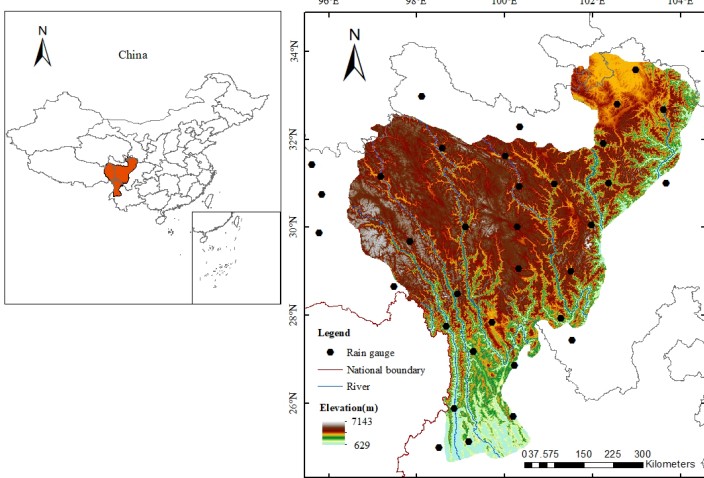

**Figure 1.** Location of study area.

*2.2. Data and Sources*

Daily rainfall data from 34 rain gauges were selected for the calculation of rainfall erosivity in the study area, ensuring a data completeness of over 98% to guarantee the accuracy of the calculations. ArcGIS 10.5 was employed to stitch and clip images to obtain the elevation map of the Hengduan Mountains region, and the zoning statistical function was utilized to extract the ground elevation points. The elevation data were sourced from Shuttle Radar Topography Mission (SRTM) with a spatial resolution of 90 m. The rainfall data used in this study were obtained from daily rainfall records provided by rain gauges, and the data were sourced from the China Meteorological Data Service Center (http://data.cma.cn/ (accessed on 23 June 2022)). To estimate potential changes in rainfall erosivity in the future, the SSP2–4.5 scenario, which closely represents future human society, from the CMIP6 dataset was selected to predict rainfall erosivity in the Hengduan Mountains region from 2015 to 2040. Considering the proximity of the study area to the southeastern Tibetan region with similar climatic conditions, four models from CMIP6 that performed well on the Tibetan Plateau were chosen for ensemble averaging [39] in predicting rainfall erosivity. To ensure comparability and informative results, daily rainfall data were selected, and statistical downscaling techniques were applied to downscale the data to the station level for statistical calculations. The specific four Global Climate Models (GCMs) used in this study are listed in the table below (Table 1).

**Table 1.** The Global Climate Model datasets used in this research.

| Model Name | Institution | Resolution (Latitude × Longitude) |
|---|---|---|
| EC-Earth3-Veg | EC-Earth-Cons (Europe) | 0.7° × 0.7° |
| MPI-ESM1.2-HR | MPI-M (Germany) | 0.93° × 0.94° |
| MRI-ESM2-0 | MRI (Japan) | 1.12° × 1.12° |
| NorESM2-LM | NCC (Norway) | 1.89° × 2.5° |

*2.3. Research Methods*

2.3.1. Calculation of Rainfall Erosivity (RE)

The Hengduan Mountains region is in the humid and semi-humid zones of China, characterized by abundant precipitation. After referring to relevant studies, this research plans to use a semi-monthly model based on daily rainfall data to calculate the rainfall erosivity. This model exhibits high accuracy with an average calculation error of 4.2%, making it suitable for the rainfall-rich southern regions. Prior to the calculation, data integrity was verified, and due to the long calculation year and large data volume, this study used Python 2.7 for batch processing of the data.

$$M_i = \alpha \sum_{j}^{k} (D_j)^{\beta} \tag{1}$$

In the equation: $M_i$ represents the rainfall erosivity of the $i$th half-month period, expressed in MJ mm ha$^{-1}$h$^{-1}$; $\alpha$ and $\beta$ are model parameters; $k$ represents the number of days in the half-month period; $D_j$ represents the daily rainfall greater than 12 mm on the $j$th day within the half-month period, while rainfall below 12 mm is considered as 0. The calculation formulas for parameters $\alpha$ and $\beta$ are as follows:

$$\beta = 0.8363 + 18.144/P(d_{12}) + 24.455/P(y_{12}) \tag{2}$$

$$\alpha = 21.586\beta^{-7.189} \tag{3}$$

In the equation, $P_{d12}$ represents the average daily precipitation with a daily precipitation ≥12 mm within the half-month period, and $P_{y12}$ represents the annual average precipitation with a daily precipitation ≥ 12 mm.

2.3.2. Calculation of Rainfall Erosivity Density (RED)

This study utilized a method of calculating rainfall erosivity density on multiple time scales. Firstly, the semi-monthly model was used to calculate the rainfall erosivity of each region on the corresponding time scale. Then, the total rainfall erosivity of each time scale was calculated and divided by the total precipitation to obtain the rainfall erosivity density. Finally, the rainfall erosivity densities calculated on different time scales were compared to analyze the different influential mechanisms of rainfall erosivity. The calculation process was mainly implemented through map algebra and can accurately reflect the rainfall erosivity situation in different regions and time scales, providing scientific basis for the management of land resources and the protection of ecological environment in the region. The calculation formula for rainfall erosivity density is as follows:

$$ED = \frac{R}{P} \tag{4}$$

where *ED* represents the rainfall erosivity density, *R* represents the rainfall erosivity, and *P* represents the precipitation.

2.3.3. The Linear Regression Model

This study used linear trend analysis to examine the trends in precipitation, rainfall erosivity, and rainfall erosivity density in the Hengduan Mountains from 1990 to 2020. Firstly, data preprocessing was conducted, including the imputation of missing values. Then, a linear regression model was used to calculate the linear trend of the data. The significance of the results was also analyzed using the *t*-test. The entire analysis process was mainly conducted using the statistical software SPSS 24. The linear regression model is expressed as follows:

$$Y = aX + b \tag{5}$$

where *Y* represents the fitted values of the linear regression, *X* represents time, *a* represents the rate of change, and *b* represents the intercept after linear fitting.

2.3.4. Theil–Sen Median Trend Analysis and Mann–Kendall Test

The trend changes in rainfall erosivity in different regions of the Hengduan Mountains were analyzed using Theil–Sen trend analysis method, and the analysis results were classified into trend grades with the Mann–Kendall (MK) test. The main calculation process was implemented using MATLAB, and the classification and visualization of the results were carried out by using ArcGIS 10.5. The calculation formula is as follows:

$$\beta_{slope} = mean\left(\frac{x_j - x_i}{j - i}\right), \forall j > i \tag{6}$$

In the equation: $\beta$ slope represents the calculated trend value, $x_i$ and $x_j$ represent the sample values of the *i*th and *j*th years, where *i* or *j* represents the time index. In this study, the change trend is defined as significant at the 0.05 confidence level, indicating significance when the value is below the 0.05 confidence level.

2.3.5. Extreme Rainfall Erosivity Index

To clarify the driving factors of rainfall erosivity, this study, based on its own characteristics and referencing the research of Wang [26], calculated four indices related to extreme rainfall erosivity for each station: Extreme Erosive Precipitation (EEP), defined as precipitation ($\geq$12 mm) exceeding the 95th percentile; Extreme Erosive Days (EED), representing the total number of days in a year with daily precipitation exceeding the threshold for extreme erosivity; Longest Continuous Erosive Days (LCED), which is the longest consecutive period of precipitation ($\geq$12 mm) meeting erosivity criteria during the study period; and Total Erosive Precipitation (TEP), representing the annual cumulative precipitation ($\geq$12 mm) meeting erosivity criteria.

## 3. Results

### 3.1. Temporal Variation Characteristics of Rainfall Erosivity in the Hengduan Mountains

#### 3.1.1. Annual Distribution of Rainfall Erosivity

From Figure 2, significant variations can be observed in both precipitation and the annual distribution of rainfall erosivity in this region. These two factors exhibit a strong correlation and synchronicity. The minimum values for both precipitation and rainfall erosivity occur in December, while the maximum values occur in July. The maximum monthly precipitation is 36.7 times higher than the minimum value, and the maximum monthly rainfall erosivity is 113.3 times higher than the minimum value. Regression analysis and correlation analysis both indicate a clear and significant positive correlation between monthly precipitation and rainfall erosivity.

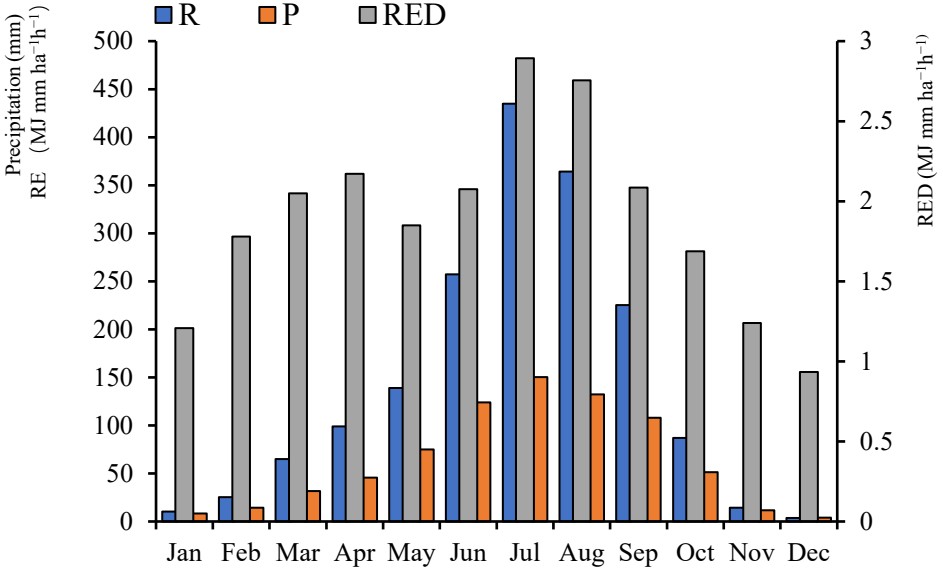

**Figure 2.** The monthly precipitation, rainfall erosivity, and rainfall erosivity density in the Hengduan Mountains region.

Both precipitation and rainfall erosivity show an increasing trend followed by a decrease. They rise from January, reach their peak in July, and gradually decrease from August to December, exhibiting distinct monsoonal characteristics. The trend of rainfall erosivity density aligns closely with that of rainfall erosivity. The average value is 1.89 MJ mm ha$^{-1}$h$^{-1}$, reflecting that single intense rainfall events are the main cause of soil erosion in this region. The rainfall erosivity density shows two peaks throughout the year, which significantly differs from the annual distribution of precipitation and rainfall erosivity. This indicates that the erosive effect of precipitation on the soil varies in different months. In months with higher rainfall erosivity, it is mainly influenced by short-duration, high-intensity rainfall events that have a destructive impact. In months with lower rainfall erosivity, the cumulative erosive effect on the soil relies more on low erosive rainfall events [40].

Influenced by the monsoon climate, the overall seasonal distribution pattern of rainfall erosivity and precipitation is as follows: summer > autumn > spring > winter (Figure 3). The coefficient of variation for seasonal rainfall erosivity was calculated to be 0.88, indicating moderate variability. On the other hand, the coefficient of variation for precipitation was 1.38, indicating strong variability. Compared to rainfall erosivity, the seasonal distribution of precipitation is more uneven, with a concentration of precipitation in summer being the main cause of soil erosion.

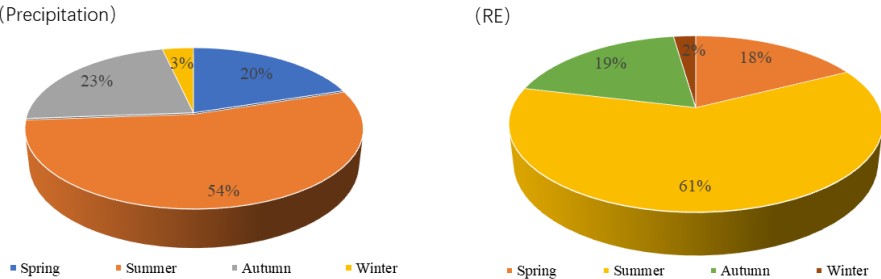

**Figure 3.** Precipitation and rainfall erosivity in different seasons.

According to the results shown in Figure 4, the pattern of rainfall erosivity density follows the order: summer > spring > autumn > winter, with values ranging from 1.47 to 2.59 MJ ha$^{-1}$h$^{-1}$. The erosivity density for all four seasons is greater than 1, indicating that the rainfall type is characterized by short duration and high intensity, which results in a higher erosive power per unit of rainfall [41]. Moreover, summer is the peak season for heavy rainfall in the region, leading to intensified cumulative soil erosion.

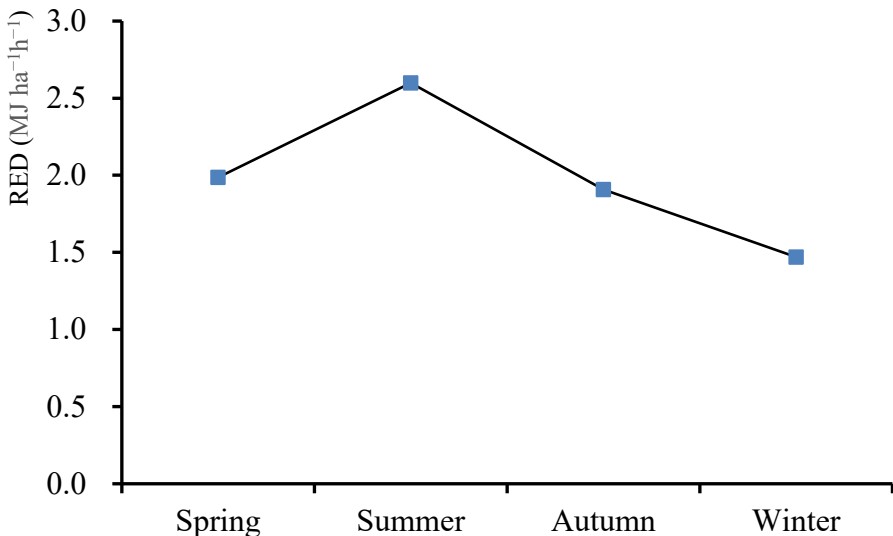

**Figure 4.** Rainfall erosivity density in different seasons.

3.1.2. Interannual Variation of Rainfall Erosivity

According to the results of linear trend analysis (Figure 5), the change rate of precipitation is −2.75 mm/10 year, indicating a non-significant decreasing trend. On the other hand, the linear change rate of rainfall erosivity is 37.43 MJ ha$^{-1}$h$^{-1}$/10 year, showing a slow increasing trend. To reduce the impact of rainfall cycles on the analysis results, a moving average was used to fit the trends of precipitation and rainfall erosivity. Based on the results of the moving average, the overall trend of precipitation during the period 1990–2001 showed a fluctuating increase, followed by a decreasing trend from 2001 to 2008, and a significant increasing trend after 2008. Similarly, the moving average results of rainfall erosivity were consistent with the changes in precipitation. From 1990 to 2001, rainfall erosivity showed a non-significant upward trend, followed by a decline from 2001 to 2009, which was statistically significant at $p < 0.05$. However, after 2009, rainfall erosivity exhibited an upward trend. Through Pearson analysis, a significant correlation was found between annual precipitation and rainfall erosivity (Figure 6), with a correlation coefficient of 0.71.

Compared to the trends in precipitation and rainfall erosivity, the rainfall erosivity density exhibits regular periodic fluctuations (Figure 5), which are also verified in the 3-year moving average. This suggests that the Hengduan Mountains region experiences years with a higher frequency of heavy rainfall in certain periodic cycles, typically around

6 to 8 years. According to the research by Panagos [40], years with higher rainfall erosivity density are more prone to flooding or drought natural disasters. By comparing the data, it is found that in the years with higher rainfall erosivity density, such as 1998, 2006, 2013, and 2020, the Hengduan Mountains region experienced various degrees of flooding or drought disasters. In 1998, a nationwide flood disaster occurred [42]. In 2006, there was a drought in the Sichuan Basin and the western plateau area [43]. In 2013, regions such as Aba and Wenchuan in western Sichuan experienced heavy rain and subsequent mudslides [44]. In 2020, the southwestern region was hit by extensive flooding due to heavy rainfall in the Yangtze River basin [42].

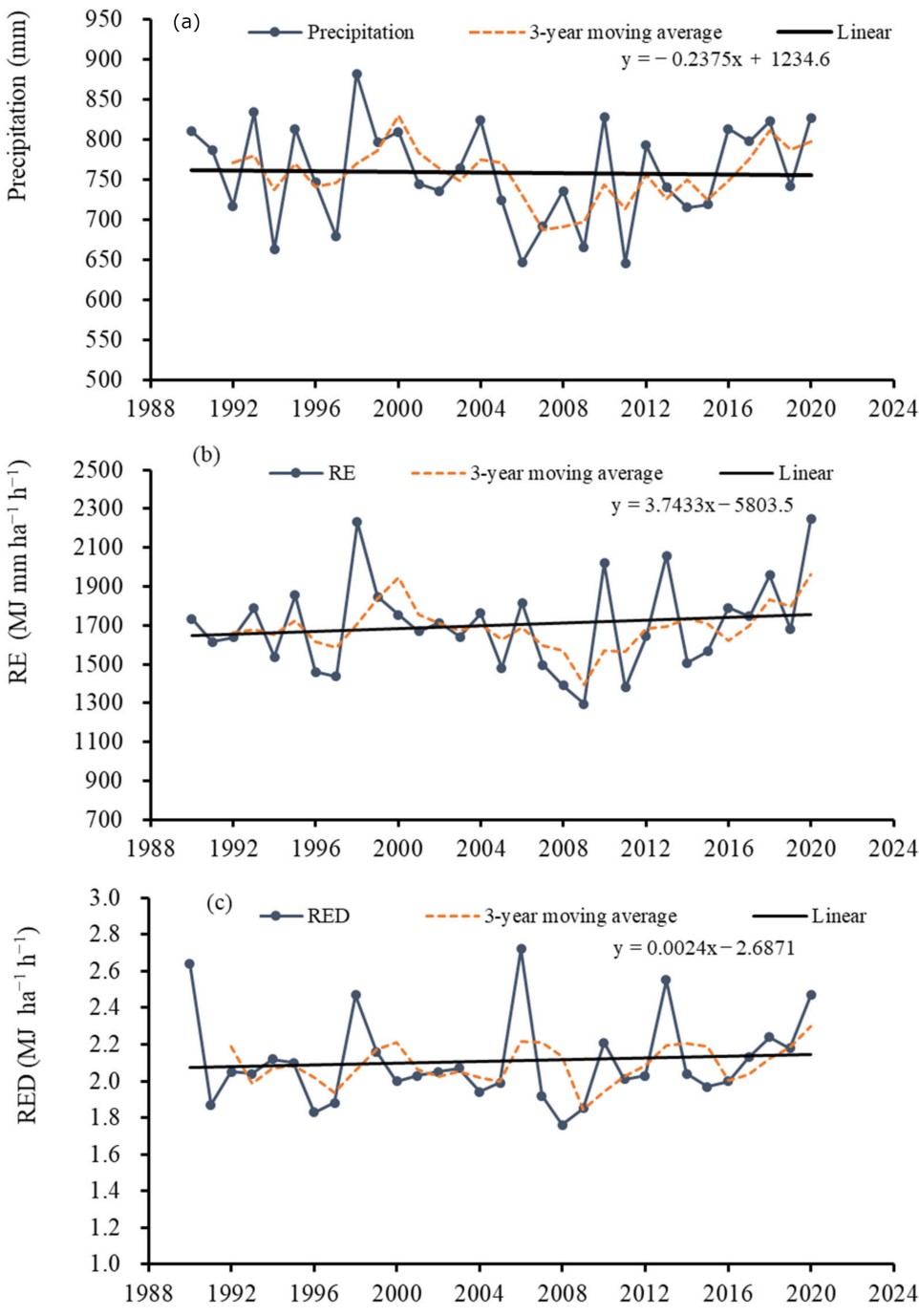

**Figure 5.** Interannual trend of (**a**) precipitation, (**b**) rainfall erosivity, and (**c**) rainfall erosivity density.

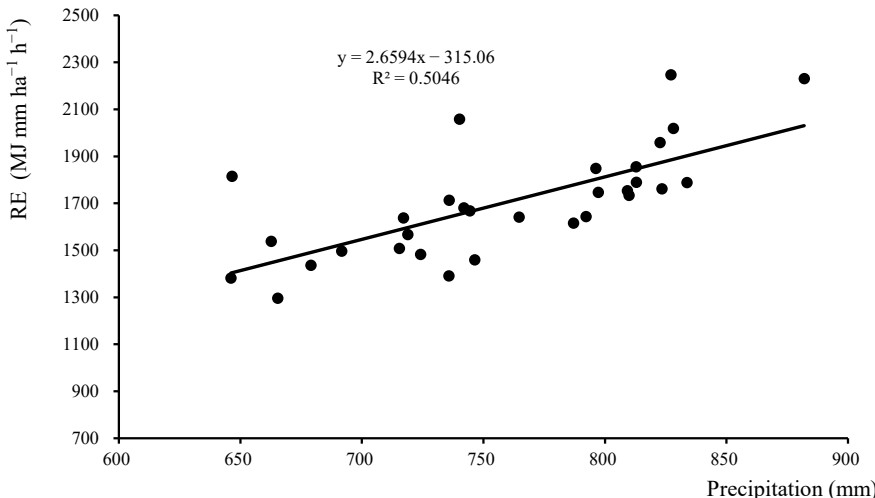

**Figure 6.** Liner relationship between annual rainfall erosivity and precipitation.

*3.2. Spatial Distribution Patterns of Rainfall Erosivity in the Hengduan Mountains*
3.2.1. Spatial Distribution Characteristics of Rainfall Erosivity in Different Seasons

Analyzing the spatial variations of rainfall erosivity enables the identification of areas prone to water erosion, as well as potential risk zones for geological disasters such as mudslides and landslides. Referring to previous studies on the spatial analysis of rainfall erosivity [45], this study employs ordinary kriging interpolation based on rainfall data collected from 34 rain gauges within the research area to analyze the spatial distribution patterns of precipitation, rainfall erosivity, and rainfall erosivity density (Figure 7).

The spatial distribution of precipitation varies among different seasons. In spring, the areas with the least precipitation are mainly located in the northwest of the Hengduan Mountains region, while the areas with the highest rainfall are primarily distributed in the southern part. In summer, the areas with the least rainfall are found in the western part of the Hengduan Mountains region, particularly in Changdu, while the areas with the highest precipitation are concentrated on the eastern edge of the Hengduan Mountains region and in the southern cities of Dali and Baoshan. During summer, due to the strengthening of the monsoon, the spatial variability of precipitation is relatively low, resulting in a more balanced spatial distribution. In contrast, in winter, which is the non-monsoon period, there is reduced moisture transport in the western part of the Hengduan Mountains region, leading to a more pronounced uneven distribution of precipitation between the eastern and western regions (Table 2).

Compared to precipitation, the spatial distribution of rainfall erosivity exhibits more regularity and a more pronounced gradient distribution. Additionally, the spatial distribution of rainfall erosivity in different seasons is influenced to varying degrees by precipitation. In spring, high values of rainfall erosivity are mainly concentrated in the southwest of the Hengduan Mountains region, while low-value areas are distributed in the northern regions of the Hengduan Mountains area. The distribution patterns of rainfall erosivity in summer and autumn are consistent with the distribution patterns of rainfall, gradually decreasing from east to west. On the other hand, in winter and spring, the distribution pattern of rainfall erosivity decreases from south to north. The spatial distribution of rainfall erosivity in spring is essentially consistent with the distribution of rainfall erosivity density. In summer and autumn, the spatial distribution of rainfall erosivity and rainfall erosivity density shows a high degree of similarity with the distribution of precipitation. In winter, the similarity between rainfall erosivity and precipitation is even higher. Therefore, it can be inferred that the distribution pattern of rainfall erosivity in this region in spring is mainly influenced by rainfall intensity. In summer and autumn, it is the combined effect of rainfall intensity and precipitation, while in winter, the spatial distribution of rainfall erosivity is primarily the result of precipitation.

In winter, the entire region exhibits low rainfall erosivity density, with the maximum value reaching only 2.27 MJ ha$^{-1}$h$^{-1}$. In contrast, the maximum value in summer reaches 6.42 MJ ha$^{-1}$h$^{-1}$. The concentrated rainfall in summer causes it to be a peak period for geological hazards during this season. Areas with steep slopes are more prone to soil erosion. In the eastern edge of the Hengduan Mountains region, not only do they experience high rainfall erosivity density during summer, but their slopes are generally above 35°. These areas are more susceptible to soil erosion and geological hazards, such as landslides and debris flows.

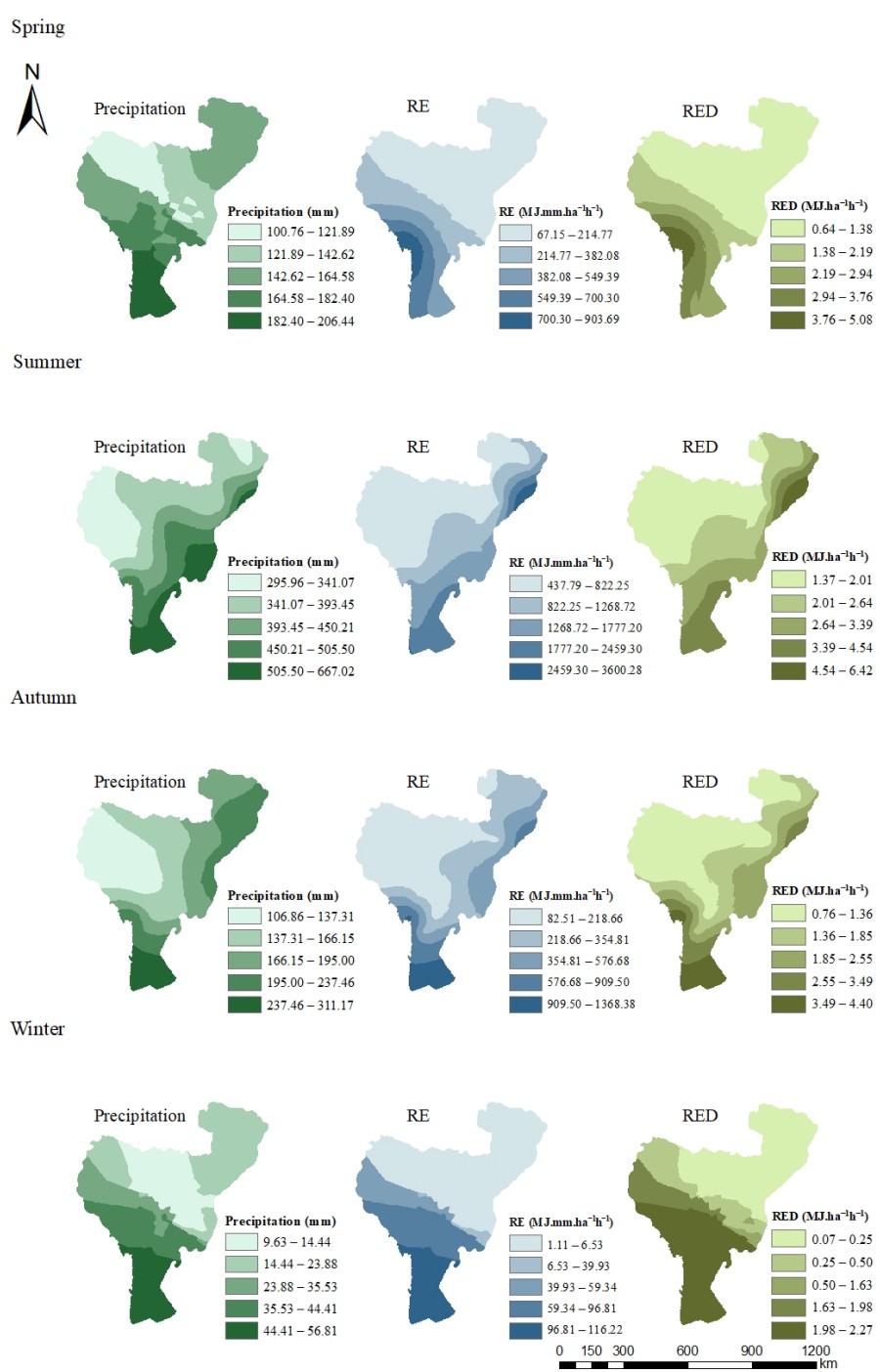

**Figure 7.** The spatial distribution of precipitation, rainfall erosivity, and rainfall erosivity density in different seasons.

**Table 2.** The spatial characteristics of seasonal average precipitation, rainfall erosivity, and rainfall erosivity density from 1990 to 2020.

| Season | Type | Min | Max | Mean | $C_v$ |
|--------|------|-----|-----|------|-------|
| Spring | Precipitation | 100.76 | 206.45 | 152.08 | 0.17 |
|  | Rainfall Erosivity | 67.15 | 903.7 | 270.96 | 0.70 |
|  | Rainfall Erosivity Density | 0.65 | 5.08 | 1.66 | 0.56 |
| Summer | Precipitation | 295.97 | 667.02 | 405.40 | 0.19 |
|  | Rainfall Erosivity | 437.79 | 3600.28 | 1045.67 | 0.52 |
|  | Rainfall Erosivity Density | 1.38 | 6.42 | 2.46 | 0.35 |
| Autumn | Precipitation | 106.87 | 311.18 | 170.84 | 0.24 |
|  | Rainfall Erosivity | 82.51 | 1368.38 | 326.46 | 0.76 |
|  | Rainfall Erosivity Density | 0.77 | 4.41 | 1.74 | 0.48 |
| Winter | Precipitation | 9.64 | 56.82 | 26.78 | 0.55 |
|  | Rainfall Erosivity | 1.12 | 116.23 | 39.53 | 1.11 |
|  | Rainfall Erosivity Density | 0.07 | 2.27 | 1.02 | 0.90 |

### 3.2.2. Spatial Distribution of Annual Average Rainfall Erosivity

By analyzing the spatial distribution maps of average annual precipitation, rainfall erosivity, and rainfall erosivity density (Figure 8), it is observed that they exhibit similar spatial distribution characteristics with significant spatial variations. They all show a decreasing gradient from southeast to northwest, consistent with the trend of the East Asian monsoon. This finding is consistent with the spatial distribution patterns of rainfall obtained by Zhang Tao [38], indicating that the spatial distribution of precipitation plays a significant role in determining the spatial variation of rainfall erosivity.

Areas with higher values of rainfall erosivity are mainly concentrated in the southern and eastern edge regions of the Hengduan Mountains, reaching up to 2500 MJ mm ha$^{-1}$h$^{-1}$. These areas receive abundant precipitation, with an average annual precipitation of 950 mm, which is consistent with the distribution of high rainfall erosivity during the summer in this region. The distribution patterns of rainfall erosivity and rainfall erosivity density are in good agreement. According to the classification results of rainfall erosivity density by Das [46], the northern part of the Hengduan Mountains region belongs to the low rainfall erosivity density category, indicating that this area experiences mainly low-intensity rainfall. In contrast, the southern part of the Hengduan Mountains region and the eastern edge region have high rainfall erosivity density.

It is worth noting that both precipitation and rainfall erosivity show a decreasing trend with increasing elevation. To further explain the influence of elevation on precipitation and rainfall erosivity, this study extracted 21,926 elevation points and their corresponding precipitation and rainfall erosivity from the Digital Elevation Model (DEM) data within the study area using ArcGIS 10.5. Through correlation analysis, it was found that elevation is significantly negatively correlated with both precipitation and rainfall erosivity, with correlation coefficients of $-0.736$ for both variables. In addition, the spatial variability of precipitation, rainfall erosivity, and rainfall erosivity density also differs. As shown in Table 3, the maximum value of precipitation is approximately twice the minimum value, while the maximum value of rainfall erosivity is about 7.5 times the minimum value.

**Table 3.** Annual variations in precipitation, rainfall erosivity, and rainfall erosivity density.

| Type | Min | Max | Mean | $C_v$ |
|------|-----|-----|------|-------|
| Precipitation | 545.73 | 1218.18 | 758.41 | 0.19 |
| Rainfall Erosivity | 644.1 | 4850.11 | 1699.93 | 0.56 |
| Rainfall Erosivity Density | 1.09 | 4.43 | 2.09 | 0.36 |

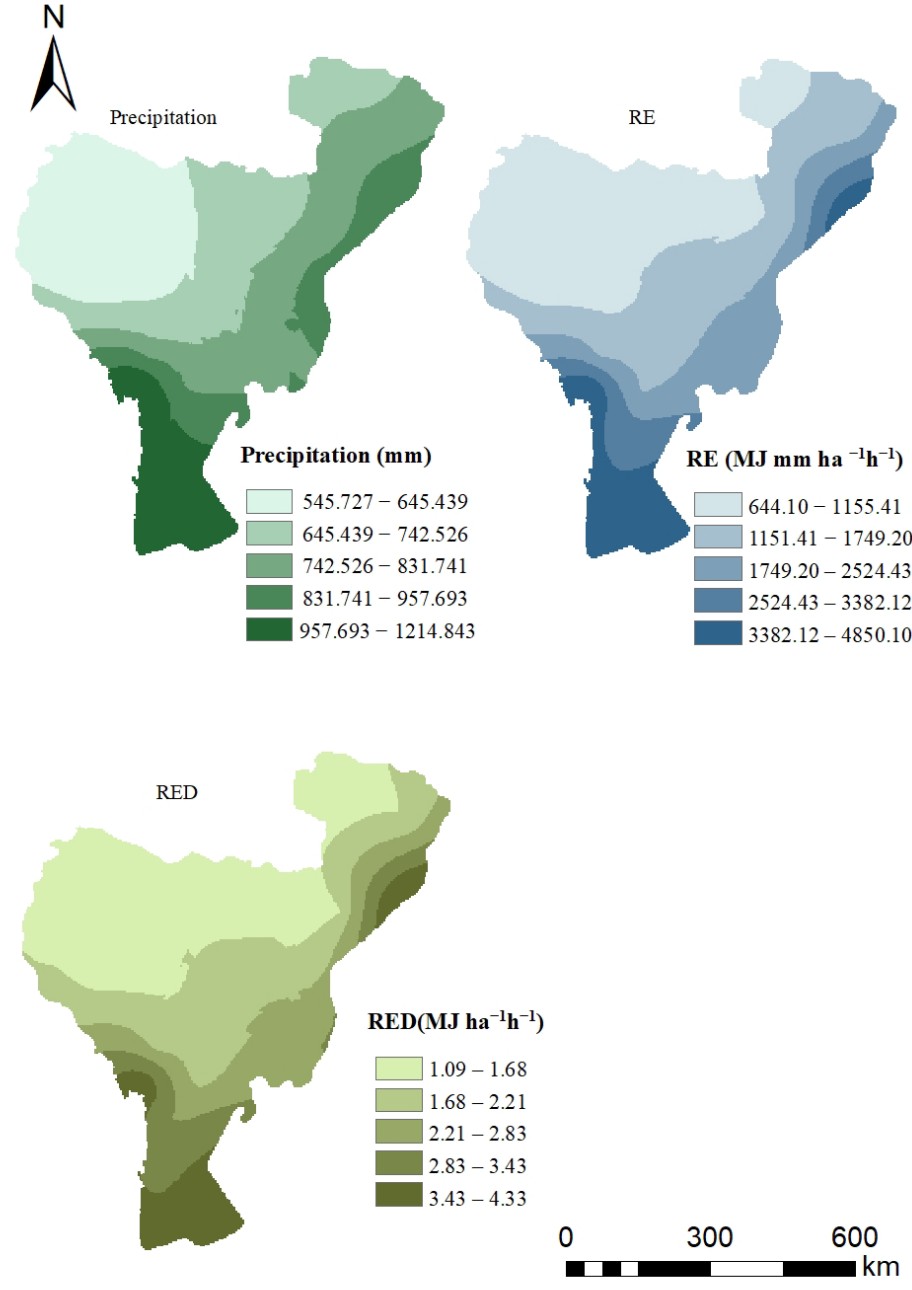

**Figure 8.** The spatial distribution of average precipitation, rainfall erosivity, and rainfall erosivity density from 1990 to 2020.

### 3.2.3. Spatial Distribution Characteristics of the Trend in Rainfall Erosivity from 1990 to 2020

In this study, the Theil–Sen trend analysis method was used to obtain the spatial distribution maps of precipitation, rainfall erosivity, and rainfall erosivity density in the Hengduan Mountains region from 1990 to 2020 (Figure 9). The results indicate significant differences in both numerical values and spatial distribution among the three variables. Overall, the rainfall erosivity shows the largest range of change, followed by precipitation, and rainfall erosivity density shows the smallest range. To enhance the reliability of the results, it is necessary to perform significance tests using the Mann–Kendall test method in combination with the obtained results.

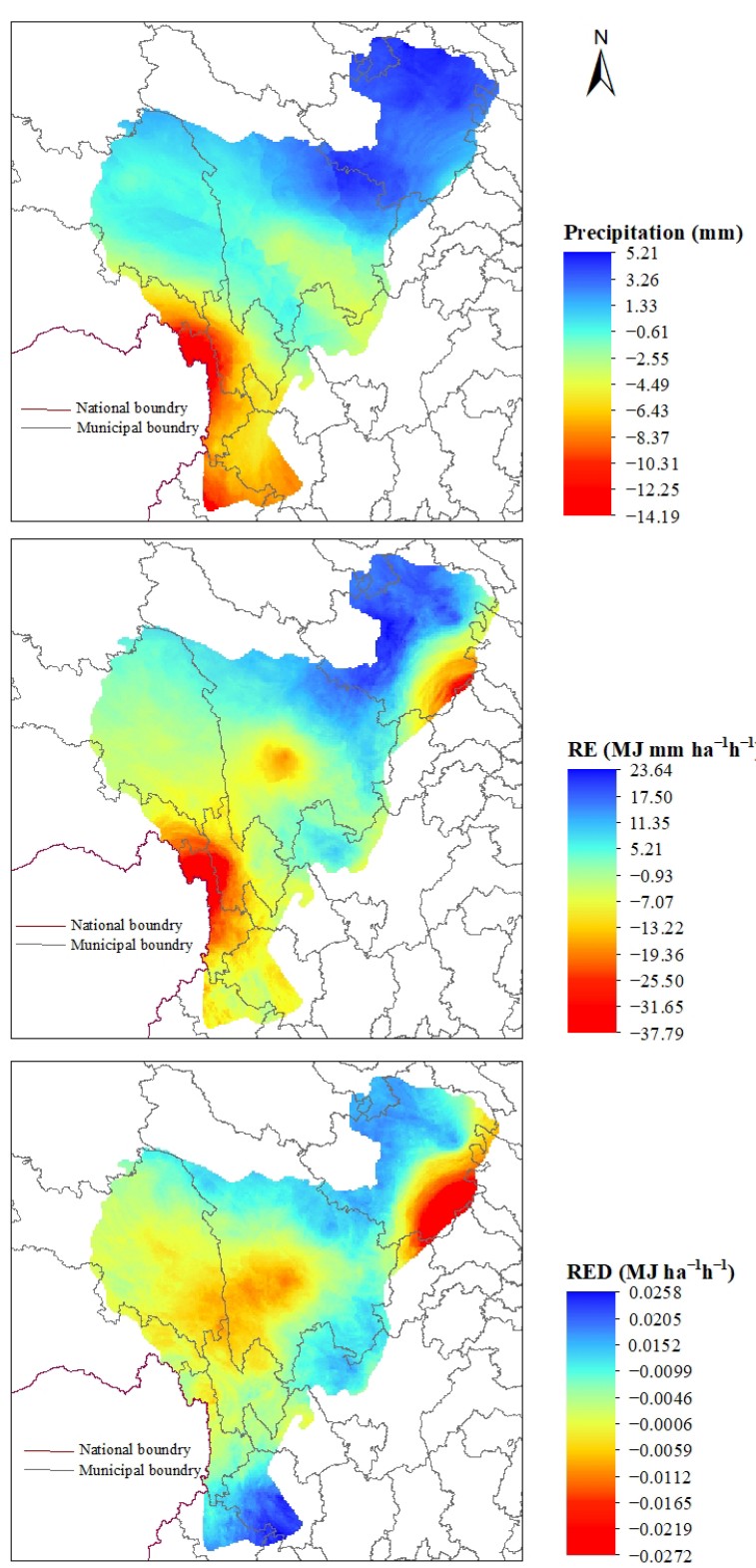

**Figure 9.** Trends in precipitation, rainfall erosivity, and rainfall erosivity density from 1990 to 2020.

Significantly increasing areas in precipitation only account for 0.18% of the study area, while significantly increasing areas in rainfall erosivity account for 6.69% of the study area (Table 4). Moreover, there is high spatial similarity between the distribution of rainfall erosivity and precipitation, indicating that the increase in rainfall erosivity is primarily attributed to the increase in precipitation. In the southern part of the Hengduan Mountains

region, there is a significant decrease in precipitation, while the northeastern part of the study area shows an upward trend. However, Xu Fei et al.'s [47] study suggests a declining trend in rainfall volume in the southern and northern parts of the Hengduan Mountains region, with a more pronounced decline in the southern part compared to the northern part. The main reason for the difference between their study and this research lies in the choice of trend analysis method and the use of different station data.

**Table 4.** Grading criteria for trend analysis.

| Type | S | Z | Trend | Area Percentage |
|---|---|---|---|---|
| Precipitation | >1 | >\|1.96\| | Significantly increase | 0.18 |
| | >1 | −1.96~1.96 | Slightly increase | 29.50 |
| | −1~1 | −1.96~1.96 | Stable | 26.65 |
| | <−1 | >\|1.96\| | Significantly decrease | 16.52 |
| | <−1 | −1.96~1.96 | Slightly decrease | 27.15 |
| Rainfall erosivity | >5 | >\|1.96\| | Significantly increase | 7.69 |
| | >5 | −1.96~1.96 | Slightly increase | 24.99 |
| | −5~5 | −1.96~1.96 | Stable | 44.31 |
| | <−5 | −1.96~1.96 | Slightly decrease | 23.01 |
| Rainfall erosivity density | >0.005 | >\|1.96\| | Significantly increase | 3.41 |
| | >0.005 | −1.96~1.96 | Slightly increase | 43.28 |
| | −0.005~0.005 | −1.96~1.96 | Stable | 41.60 |
| | <−0.005 | −1.96~1.96 | Slightly decrease | 11.7 |

Slope value of S: Theil–Sen method; trend value of Z: Mann–Kendall method.

The comparison of the significance distribution maps for precipitation, rainfall erosivity, and rainfall erosivity density (Figure 10) indicates a decreasing trend in precipitation and rainfall erosivity in the southern and eastern edge regions of the Hengduan Mountains, where precipitation is relatively high. The main reason for this trend is the weakening of the East Asian summer monsoon in the 1970s, which resulted in reduced rainfall in the southern region [48]. In comparison to the variations in precipitation, both rainfall erosivity and rainfall erosivity density exhibit greater stability. The stable areas account for more than 40% of the total area, while the areas with unchanged precipitation represent 26.65% (Table 4). The regions where rainfall erosivity and rainfall erosivity density remain stable are generally consistent with the areas of stable precipitation within the region.

*3.3. Prediction of Rainfall Erosivity in the Hengduan Mountains*

3.3.1. Prediction of Rainfall Erosivity Index

For the prediction of rainfall erosivity, this study mainly used data from four Global Climate Models (GCMs) in CMIP6. To provide reference and facilitate comparative analysis, the reference period was set from 2005 to 2020. By comparing the rainfall erosion indices between the reference period and forecast period (2025–2040), the changes in future rainfall erosivity in the Hengduan Mountains were analyzed.

According to the statistical results in Table 5, it can be observed that, compared to the reference period, 76% of the stations show a decreasing trend in extreme erosive precipitation (EEP). The stations with the largest decrease in extreme erosive precipitation are mainly located in the eastern edge of the Hengduan Mountains and regions with abundant precipitation, such as Yunnan. Conversely, the stations with an increase in extreme erosive rainfall are mainly found in areas with less precipitation, such as the Ganzi and Aba regions. The number of extreme erosive days (EED) and the total annual erosive precipitation (TEP) for the selected stations both exhibit an increasing trend. The longest consecutive erosive days (LCED) also show an increase, except for the Daofu station. The regions with the most significant increase in total annual erosive precipitation are mainly concentrated in areas with less precipitation and lower rainfall erosivity, such as Ganzi, Aba, and Changdu.

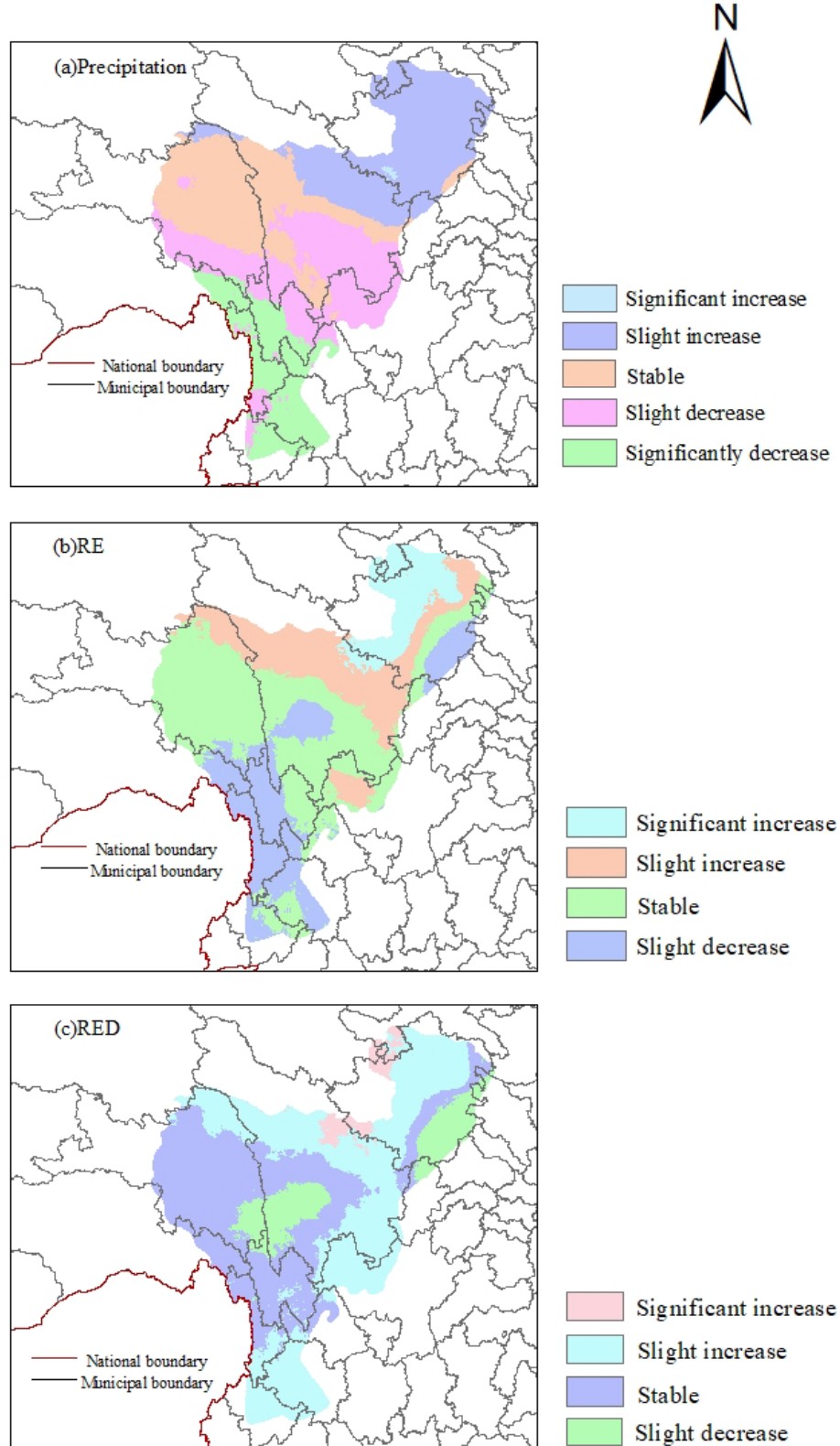

**Figure 10.** Spatial distribution of precipitation, rainfall erosivity, and rainfall erosivity density trend grades.

**Table 5.** Rainfall erosivity index during reference and forecast period.

| Rain Gauge | Reference Period (2005–2010) | | | | Forecast Period (2025–2040) | | | |
|---|---|---|---|---|---|---|---|---|
| | EEP | EED | LCED | TEP | EEP | EED | LCED | TEP |
| Shiqu | 25.8 | 1 | 3 | 179.9 | 21.4 | 1 | 6 | 276.5 |
| Dingqing | 30.3 | 1 | 4 | 245.1 | 23.7 | 1 | 7 | 290.3 |
| Luolong | 27.2 | 1 | 2 | 129.0 | 26.7 | 1 | 8 | 384.0 |
| Bomi | 37.5 | 1 | 6 | 397.9 | 34.4 | 3 | 11 | 882.6 |
| Chayu | 39.9 | 1 | 4 | 362.4 | 42.3 | 3 | 14 | 1335.8 |
| Gongshan | 50.5 | 2 | 7 | 1065.6 | 39.3 | 3 | 13 | 957.8 |
| Nujiang | 50.2 | 1 | 4 | 577.2 | 37.0 | 3 | 16 | 834.4 |
| Tengchong | 49.4 | 2 | 5 | 1001.0 | 35.8 | 3 | 16 | 797.5 |
| Baoshan | 53.3 | 1 | 5 | 589.4 | 34.8 | 2 | 11 | 697.1 |
| Yanyuan | 41.5 | 1 | 3 | 467.0 | 34.2 | 2 | 11 | 661.0 |
| Jiulong | 30.8 | 1 | 4 | 481.3 | 31.5 | 2 | 11 | 702.2 |
| Kangding | 32.5 | 1 | 4 | 422.9 | 33.5 | 2 | 8 | 795.7 |
| Dujiangyan | 90.8 | 1 | 6 | 790.3 | 41.9 | 2 | 8 | 707.7 |
| Songpan | 29.5 | 1 | 4 | 307.4 | 37.0 | 2 | 8 | 651.1 |
| Ruoergai | 33.2 | 1 | 3 | 308.5 | 30.1 | 2 | 6 | 538.1 |
| Hongyuan | 29.3 | 1 | 3 | 342.1 | 30.7 | 2 | 7 | 623.1 |
| Seda | 27.6 | 1 | 4 | 289.3 | 24.0 | 1 | 7 | 405.3 |
| Dege | 28.9 | 1 | 4 | 278.9 | 23.1 | 1 | 7 | 369.0 |
| Ganzi | 26.1 | 1 | 3 | 282.5 | 24.5 | 2 | 9 | 467.9 |
| Changdu | 34.4 | 1 | 3 | 164.0 | 24.8 | 1 | 8 | 309.7 |
| Xinlong | 29.1 | 1 | 5 | 266.4 | 26.3 | 2 | 9 | 568.9 |
| Daofu | 30.7 | 1 | 9 | 288.1 | 28.2 | 2 | 7 | 670.8 |
| Zuogong | 27.7 | 1 | 3 | 178.4 | 29.8 | 1 | 7 | 453.7 |
| Batang | 30.4 | 1 | 4 | 178.3 | 26.2 | 1 | 10 | 378.7 |
| Litang | 32.9 | 1 | 4 | 344.0 | 27.1 | 2 | 9 | 604.2 |
| Daocheng | 31.7 | 1 | 5 | 309.1 | 28.5 | 2 | 11 | 630.8 |
| Deqin | 37.5 | 1 | 5 | 281.2 | 32.0 | 2 | 11 | 747.1 |
| Shangri-La | 37.1 | 1 | 3 | 287.5 | 31.3 | 2 | 11 | 693.1 |
| Muli | 42.3 | 1 | 4 | 474.0 | 33.8 | 2 | 13 | 697.6 |
| Weixi | 40.0 | 1 | 5 | 517.0 | 34.7 | 3 | 13 | 756.3 |
| Maerkang | 32.1 | 1 | 5 | 408.8 | 31.1 | 2 | 7 | 736.7 |
| Xiaojin | 29.3 | 1 | 4 | 249.4 | 35.9 | 3 | 12 | 1589.3 |
| Dali | 57.2 | 1 | 4 | 708.5 | 33.0 | 2 | 10 | 1076.7 |
| Lijiang | 50.0 | 1 | 4 | 580.1 | 31.5 | 2 | 10 | 1156.7 |

### 3.3.2. Comparison of Variation in Rainfall Erosivity between Reference and Forecast Period

After conducting a statistical analysis of the rainfall erosivity for all stations during the reference period (2005–2020) and the forecast period (2025–2040), it was found that 85% of the stations showed a significant increase in rainfall erosivity according to the forecast results. However, in areas with abundant precipitation and high rainfall erosivity, such as Dujiangyan, Dali, and Lijiang, the rainfall erosivity show a decreasing trend (Figure 11), but the precipitation of these sites has been increasing. Combined with the results in Table 4, it can be seen that the extreme erosive precipitation (EEP) of these sites has decreased significantly. This further confirms that the reason for the decrease in rainfall erosivity is mainly due to the decrease in single rainfall intensity, based on the results shown in Figure 11. However, according to the results of outlier analysis, the occurrence of extreme erosion condition is expected to decrease, but the majority of stations will see an increase in extreme rainfall erosivity values. Specific to extreme erosion events, the sites of Dujiangyan, Shangri-La, and Dali are expected to experience a decrease in their extreme erosion values compared to the reference period. Conversely, extreme rainfall erosivity values for stations in the Yunnan region are expected to increase. Extreme erosion events are therefore more likely to occur in the Yunnan region.

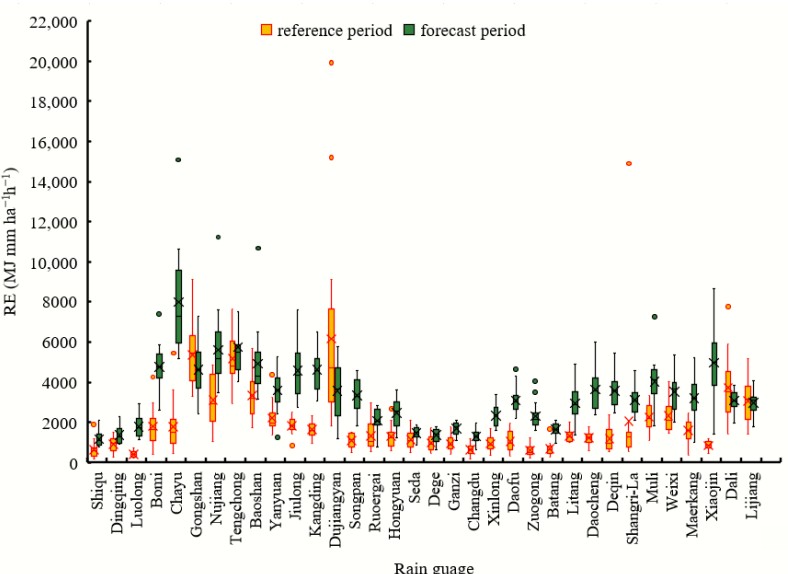

**Figure 11.** Annual rainfall erosivity for each rain gauge during the reference and forecast periods. Comparison of annual rainfall erosivity during the reference and forecast periods. The "×" represents the average value of rainfall erosivity during the study period, while "·" represents the outlier values of the site during the study period.

To further analyze the changes in rainfall erosivity under the SSP2–4.5 scenario, a comparison was made between the forecast period and reference period for rainfall erosivity and rainfall erosivity density. This analysis resulted in a spatial distribution map of rainfall erosivity variation (Figure 12). According to the spatial distribution of changes in rainfall erosivity and rainfall erosivity density, only 1.17% of the entire Hengduan Mountains region showed a decrease in rainfall erosivity compared to the reference period. This indicates that the issue of soil erosion in the future will become even more severe. However, in most areas, the rainfall erosivity density decreases, covering 73.3% of the study area. This reflects the reduction in the influence of rainfall intensity on rainfall erosivity, with erosion being driven more by continuous rainfall. The regions with the most significant increase in rainfall erosivity are mainly located in the western edge of the Hengduan Mountains, and these areas also experience a significant increase in rainfall erosivity density. Consequently, these regions will face a greater threat of soil erosion caused by heavy rainfall in the future.

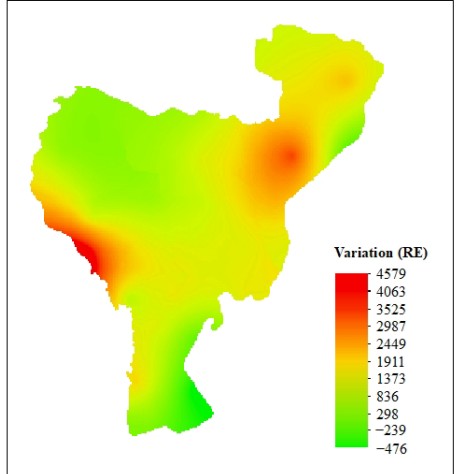
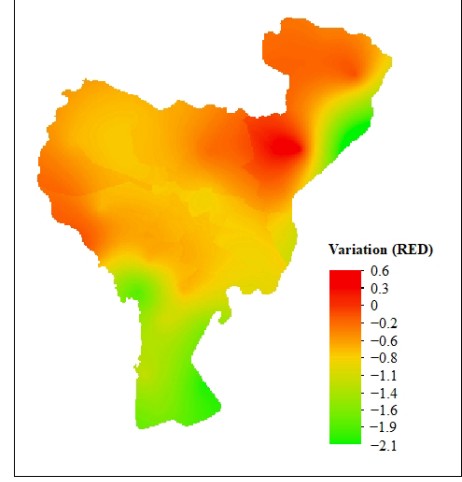

**Figure 12.** The variation in forecast of rainfall erosivity and rainfall erosivity density compared with the reference period.

## 4. Discussion and Conclusions

### 4.1. Discussion

After conducting a statistical analysis of the intra-annual distribution of rainfall erosivity in the Hengduan Mountains region, it is evident that the rainfall erosivity in this area exhibits typical monsoon characteristics. This is manifested not only in the significant difference between the rainfall erosivity values during the monsoon and non-monsoon periods but also in the varying spatial distribution patterns of rainfall erosivity in different seasons, influenced by the amount of precipitation. According to the research by Liang Jingyu et al. [49], the annual distribution pattern of rainfall erosivity in this region is closely related to the radiative effect of the summer subtropical high-pressure system. Furthermore, they pointed out that as rainfall intensity increases, the erosion density also increases, further indicating that, in areas with high rainfall erosivity density, rainfall erosivity is primarily driven by highly erosive heavy rainfall.

The precipitation erosivity in the Hengduan Mountain region showed a decreasing trend from southeast to northwest, and this variation is closely related to the topographical features of the region. The mountainous terrain runs from south to north, with lower elevations in the eastern and southern parts. The mountain range acts as a barrier, blocking the moisture transported by the monsoon and causing it to accumulate in the eastern and southern parts of the Hengduan Mountains region [50]. Moreover, these areas are characterized by abundant vegetation and forests, resulting in high annual precipitation and subsequently high rainfall erosivity. However, as the elevation increases, the north–south mountain range blocks the monsoon, leading to thinner moisture above 1500 m and predominantly grassland vegetation types with less precipitation, resulting in the lowest rainfall erosivity.

In the eastern edge of the Hengduan Mountains region and the southern region, where rainfall erosivity is high and intense rainstorms are prone to occur in summer, a study conducted by Sun [51] found that these regions were highly susceptible to flood disasters. Moreover, coupled with steep slopes and significant surface undulations, these areas were also prone to debris flows and flash floods, as revealed in the study by Hu Kaiheng et al. [52]. Therefore, it is speculated that rainfall erosivity plays a significant triggering role in geological hazards. Thus, in soil erosion control, attention should be given not only to areas requiring priority management but also to the prevention of secondary disasters caused by soil erosion. Hence, this study has practical implications for soil erosion protection work.

Under global climate warming and human influence, significant differences are observed in the increase or decrease of precipitation in different areas of the Hengduan Mountains region, resulting in varying trends of rainfall erosivity across regions. From the perspective of rainfall erosivity prediction, although extreme erosive rainfall is expected to decrease in the future, the combined effects of erosion frequency and continuous erosion duration indicate that the Hengduan Mountains region will face more severe soil erosion issues, particularly with high-frequency and high-intensity long-duration erosions. In a study on soil erosion prediction in the Qinghai-Tibet Plateau by Teng Hongfen et al. [53], it was found that rainfall erosivity will increase in the future, leading to intensified soil erosion. Since the Hengduan Mountains region is geographically adjacent to the Qinghai-Tibet Plateau, this prediction implies that future climate change poses severe challenges to soil management in China. The widespread increase in rainfall frequency will elevate the risk of geological hazards in many areas of China and significantly impact agriculture.

This study provides a comprehensive analysis of the temporal and spatial scales of rainfall erosivity changes, elucidates the distribution mechanism of rainfall erosivity in this region, and effectively mitigates the influence of anomalous years on trend analysis by employing Theil–Sen trend analysis and the MK test. It visually depicts the degree of change in precipitation, rainfall erosivity, and rainfall erosivity density in the Hengduan Mountains region from 1990 to 2020.

This study also has certain limitations and shortcomings. The rainfall data corrected by the Delta method were utilized, and the rainfall erosivity was predicted through ensemble averaging. This approach addressed the issue of low accuracy in large-scale data and compensated for the limitations of single climate models in simulating accuracy, significantly enhancing the reliability of the predictions. However, it is important to note that the CMIP6 data used in this study inherently exhibit systematic biases towards wetness [54], which can affect the accuracy of the results and, to some extent, lead to an overestimation of rainfall erosivity. Additionally, due to the study area's location in the southwestern border region of China, there is a scarcity of ground stations compared to the eastern coastal areas. Moreover, each station covers a limited area, resulting in limited accuracy of the interpolated results.

The causes of soil erosion are multifaceted, and vegetation, topography, and rainfall can all influence the extent of soil erosion [55]. This study analyzed the variation patterns of rainfall erosivity, providing a certain basis for soil erosion control in this area. However, how to enhance the accuracy of data and adopt more precise research methodologies to compensate for the limitations of our current study will be the focus of future research efforts.

### 4.2. Conclusions

This article explored the spatiotemporal changes of rainfall erosivity in the Hengduan Mountains and identified an increasing trend in the region in recent years, with the most significant increase observed in the northeast. The rainfall intensity and erosivity density in this region have also increased. The rainfall erosivity trend decreases from southeast to northwest, with the southern and eastern regions being more susceptible to water erosion. The forecast suggests that the erosive rainfall events and duration in the Hengduan Mountains will continue to increase in the future, exacerbating soil and water loss, which calls for strengthened management efforts.

**Author Contributions:** Conceptualization, X.L.; investigation, data curation, and the main modification of works, L.Z.; methodology, X.L., L.Z. and S.H.; project administration, K.S. and Z.Z.; resources, Z.Z.; software, validation, and visualization, L.Z.; writing—original draft, L.Z.; writing—review and editing, X.L., S.H. and Z.Z. All authors have read and agreed to the published version of the manuscript.

**Funding:** This study was funded by the National Key Research and Development Program (2022YFF1302903).

**Data Availability Statement:** Data are available from the authors upon request.

**Conflicts of Interest:** The authors declare no conflict of interest.

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
