# Peer review of "Characteristics and Projection of Rainfall Erosivity Distribution in the Hengduan Mountains"

_land, doi:10.3390/land12071435_

Round 1
Reviewer 1 Report
This manuscript investigated the temporal and spatial variations of rainfall erosivity in the Hengduan Mountains and predicted its future trends. In general, this manuscript provided ample and multidimensional exploration on the rainfall erosivity in the Hengduan Mountains, with certain practical significance for controlling soil erosion and water loss in the region. I believe that this manuscript would be of great interest to the readers of Land journal.
Overall, the paper meets the requirements of the Journal, and is worthy of publication. Before accepting for publication, please address the following minor revisions:
The abstract is well written and captivating. However, the article is too long, it should be cut down. Furthermore, the conclusion part should be re-summarized using more concise expressions.
Line 189-191: This paragraph is not related to the research methods author introduced.
The author discussed the influence of altitude on the spatial distribution of rainfall erosivity in the results section. It is necessary to reiterate this point in the discussion section.
The article predicted the rainfall erosivity in the Hengduan Mountains based on CMIP6 data. Please state the function of CMIP6 for this study in detail.
Line 310-312: It is not relevant to the research.
Line 317-318: It is not relevant to the research.
Line 397-398: Please provide a detailed explanation of how the author extracted the data related to altitude.
Line 491-492: Please use percentages to describe the increase in rainfall erosivity at the relevant sites instead of “most”, which would be more accurate.
Line541-542: This sentence does not seem appropriate in this part.
Please consolidate the limitations and shortcomings of the research in the discussion section.
Minor editing of English language required
Reviewer 2 Report
Row 71- surpassing the national average and making it one of the most severely eroded regions in China.the national average and making it one of the most severely eroded regions in China.- Error!
Row 280 - Panagos P [36] – without P !
Row – 480 Table 4- What are S and Z – please give information.
The manuscript is too long, it can be reduced. Some parts of the manuscript are too similar - written in different ways.
The Universal Soil Loss Equation (USLE) or RUSLE can be used to predict the long-term average annual rate of soil erosion. These methods are well-known and reliable.
Zhang Wenbo 2002- method is used mainly in China, but it is not so popular.
Reviewer 3 Report
Dear Authors,
The manuscript is about a very important topic.
However, I see that you need a much stronger and more elaborated introduction.
The materials and methods part need some strengthening as well. Make it more understandable, especially from the point of view of concentrating on the methods and materials, and try to clear the parts belonging to the introduction part as much as possible.
The results section also includes some literature review/discussion that should be separated.
The figures and chapter titles/subtitles are not good enough. These parts need a lot of re-working and making them self-explanatory.
I think with a little bit of effort and work the manuscript can be reconsidered for publication. It is easier to reject and restart than re-work. I think.
Regards, Reviewer X
PS: see my detailed comments as examples in the pdf! Please apply them to the corresponding parts, I did not do all the commenting all over the text as these are typical errors, I think.

I think English is basically OK, there are some minor mistakes/errors.
Round 2
Reviewer 1 Report
no more questions.